# The Influence of Starch Modification with Amylosucrase Treatment on Morphological Features

**Hyeyoung Lee [1] and Inmyoung Park [2],***

[1]   Division of Applied Bioengineering, Dong-eui University, Busan 47340, Korea; hlee@deu.ac.kr
[2]   Division of Food and Culinary Arts, Youngsan University, Busan 48015, Korea
*   Correspondence: inmpark@ysu.ac.kr; Tel.: +82-51-540-7236

**Abstract:** Amylosucrase (AS) is a starch-modifying enzyme from *Neisseria polysaccharea* used to produce low-glycemic starches such as slowly digestible starch (SDS) and resistant starch (RS). The morphology of native, control, and AS-modified waxy corn starches (230 and 460 U) was examined using a particle size analyzer and field-emission scanning electron microscopy (FE-SEM). AS modification of the starch elongated the glucose and resulted in higher SDS and RS contents. The mean particle sizes of the control, 230 U-AS-, and 460 U-AS-treated starches were 56.6 μm, 128.0 μm, and 176.5 μm, respectively. The surface of the 460 U-AS-treated starch was entirely porous and coral-like, while the 230 U-AS-treated starch had a partial dense and flat surface which did not react with AS. FE-SEM of the granule cross section confirmed that the center contained a dense and flat region without any evidence of AS reaction to either of the AS-treated starches. It was assumed that the particle size and porous and sponge-like particle features might be related to the SDS and RS fractions.

**Keywords:** amylosucrase-modified starch; particle size distribution; cross-sectioned particle; morphology; low glycemic starch

## 1. Introduction

Starch is the major dietary source of carbohydrates and plays an important role in energy supply in humans. It is made up of a large number of glucose units, amylose and amylopectin, linked by $\alpha$-d-(1→4) and/or $\alpha$-d-(1→6) glycosidic bonds. Starch is classified into rapidly digestible starch (RDS), slowly digestible starch (SDS), and resistant starch (RS) based on the rate of glucose release and its absorption in the intestine [1]. RS is indigestible in the small intestine but is fermented in the large intestine by colonial bacteria into short-chain fatty acids. SDS is completely digested in the small intestine through a slow digestive process and is the main determinant of the postprandial glucose level [2,3]. Both SDS and RS have a low glycemic index (GI) and thereby reduce the risk of developing diabetes, obesity, and cardiovascular diseases [3,4].

Amylosucrase (AS, E.C. 2.4.1.4) from *Neisseria polysaccharea* (NPAS) has gained attention due to its ability to generate starches with high amounts of low-glycemic-index starch fractions SDS and RS. AS uses sucrose as a substrate and elongates the non-reducing end of amylopectin and amylose external chains by up to 13–19 glucose units by its transglycosylation activity when incubated with sucrose at 37 °C [5–7]. NPAS treatment extends the branch chains in amylopectin and amylose, generating more stable double helices that are further stabilized by hydrogen bonds, leading to an altered distribution of the crystalline region and resulting in decreased digestibility and increased SDS and RS content [8–13]. Previous studies using Fourier-transform infrared spectroscopy (FT-IR) and cross-polarization magic angle spinning (CP/MAS) $^{13}$C-NMR techniques demonstrated that AS-treated starch has altered side chain length distribution and molecular crystalline structures [10].

Potoki-Veronese et al. [6] synthesized amylose *in vitro* with NPAS using sucrose as the only substrate, without using any primers. They used different initial sucrose concentrations (100, 300, and 600 mM) and observed the morphological differences between the products. An increase in sucrose concentration resulted in an increase of maltooligosaccharides and amylose chains and also a homogeneously rounder and bigger morphology. AS-treated starch morphology data have been published for various reaction times and acid treatments after AS treatment, and for pre-gelatinized starch [14–17]; however, the morphology of entirely finished reaction products with different AS treatments and the inner morphological structures of AS-treated starch have not yet been published, to our knowledge.

The aim of this study was to observe and characterize the surface and inner morphological features of starch prepared using two different NPAS concentrations (230 U and 460 U) via field-emission scanning electron microscopy (FE-SEM) and to compare them with those of native and control starch; this might provide a better understanding of the action pattern of glucose elongation in starch using AS. Also, we aimed to identify how the particle size and morphology of starch are related to its nutritional fractions (RDS, SDS, and RS).

## 2. Materials and Methods

### 2.1. Experimental Materials

Waxy corn starch was obtained from MSC Co. (Yangsan City, Korea). Sucrose, pancreatin (P7545, activity 8 × USP/g), and amyloglucosidase (A7095, AMG 50 mL, activity 300 AGU/mL) were obtained from Sigma Chemical Co. (St. Louis, MO, USA). All chemicals were of analytical grade. NPAS (amylosucrase from *Neisseria polysaccharea*) was prepared as previously described [8].

### 2.2. Preparation of AS-Treated Starch

Enzyme-modified starch was prepared as described in a previous study [10]. A starch suspension (5%, w/w) was prepared by mixing starch with 100 mM sodium citrate buffer (pH 7.5) containing 100 mM sucrose and made up to a final volume of 30 mL. The suspension was boiled for 30 min. After cooling to 30 °C, AS (230 or 460 U) was added to the suspension and incubated in a water bath at 30 °C for 20 h. The reaction was terminated by adding three volumes of 70% ethanol. The AS-treated starch suspension aggregated with the 70% ethanol addition. The suspension of aggregated AS-treated starch was precipitated by centrifugation at 10,000× *g* for 10 min, and the supernatant was removed. The precipitate was washed three times with distilled water and centrifuged at 10,000× *g* for 10 min to remove the remaining ethanol and the soluble fraction. The pellet was freeze-dried, ground, and sieved using a 70-mesh screen (0.210 mm sieve opening). Control starch was prepared following the same procedure as for AS-treated starch, except for AS addition.

### 2.3. Particle Size Analysis

The particle size distribution was measured using a Beckman Coulter LS 13 320 Laser Diffraction Size Analyzer (Beckman Coulter, Inc. Brea, CA, USA) equipped with a polarization intensity differential scattering (PIDS) detector with tungsten–halogen lamp. Native and AS-treated starch samples were suspended in 94% ethanol (Duksan Pure Chemical Co., Ltd., Ansan City, Korea) and introduced into the universal liquid module until an obscuration of 8–12% was achieved using 94% ethanol as a fluid [18]. No swelling was observed during measurements. The control starch suspension was sonicated for 10 min to ensure that the suspension had not aggregated. Where, as a preliminary experiment, the sonication time was determined as the time when particles no longer separated and the unmodified particle size distribution was made using the particle size analyzer. AS-treated starch did not require sonication because no change in particle size distribution was observed without sonication.

## 2.4. Field-Emission Scanning Electron Microscopy (FE-SEM)

The surface structures and the cross section of AS-treated starches (230 U and 460 U) were observed using FE-SEM. Starch samples were mounted on round aluminum stubs covered with double-sided adhesive tape and coated with gold using a sputter coater (E-1010, Hitachi Ltd., Tokyo, Japan) with a 15 mA discharge current under vacuum. They were then examined via FE-SEM (Supra 25, Carl Zeiss, Oberkochen, Germany) at an accelerating voltage of 10 kV. The cross-sectioning of the AS-treated starch was performed following Lee et al. [19]. Briefly, the AS-treated starch particles were cross-sectioned using a stainless-steel double razor blade of about 2 μm thickness (ST-300, Dorco, Seoul, Korea). Magnifications varied between 400× and 2000×.

## 2.5. Statistical Analysis

The data are reported as the means of triplicate measurements with standard deviations. Analysis of variance (one-way ANOVA) was performed, and differences in the means of samples were analyzed via Duncan's multiple range tests ($p < 0.05$) using SPSS Statistics (ver. 25.0, IBM Corp., Armonk, NY, USA).

## 3. Results and Discussion

### 3.1. Particle Size Distribution

Using a particle size analyzer, we determined that the average particle size of waxy corn starch was 32.3 μm and that of control starch was 56.6 μm. The measured size of the waxy corn starch coincided with that in previous reports [18] and the FE-SEM images (Figure 1). The control starch showed increased particle size because of water swelling during gelatinization. After AS treatments, the particle size of the 230 U-AS-treated starch was more than 2 times greater (128.0 μm) and that of 460 U-AS-treated starch was 3 times greater (176.5 μm) than that of control starch (56.6 μm) (Table 1). Detailed information on the d10 (the particle diameter of the particle that 10% of the cumulative sample mass), d50 (the particle diameter of the particle that 50% of the cumulative sample mass), and d90 (the particle diameter of the particle that 90% of the cumulative sample mass) values of 230 U- and 460 U-AS-treated starch is shown on Table 1. These results support the statement that AS elongated glucose at the non-reducing end of the starch molecules, thereby increasing the molecular weight ($M_w$) and radius of gyration ($R_z$) of AS-treated starch. When 40,000 U of AS was added to waxy rice, the $M_w$ approximately doubled and $R_z$ increased by 1.4 times, with the variation depending on the starch type, amount of enzyme treated, reaction time, and so on [12]. During maltooligosaccharide synthesis with AS treatment in the early stage of experiments, AS reacted with higher amounts of substrates (sucrose) to produce a larger diameter, more homogeneous and individual ovoidal shape [6].

**Table 1.** Particle sizes of waxy corn starch treated with amylosucrase (AS).

|  | d10 [3] | d50 [3] | d90 [3] | Mean |
|---|---|---|---|---|
| Native | 1.2 ± 0.01 [1,4,5] | 26.2 ± 0.11 [1] | 60.3 ± 0.85 [1] | 32.3 ± 0.66 [1] |
| Control [1] | 5.5 ± 0.02 [2] | 43.2 ± 0.24 [2] | 130.7 ± 0.33 [2] | 56.6 ± 0.06 [2] |
| 230 U-AS [2] | 43.9 ± 1.04 [3] | 129.5 ± 0.21 [3] | 207.3 ± 0.84 [3] | 128.0 ± 0.58 [3] |
| 460 U-AS | 67.7 ± 0.59 [4] | 170.1 ± 0.54 [4] | 293.4 ± 1.35 [4] | 176.5 ± 0.37 [4] |

[1] Control for AS-treated starches; [2] 230 U and 460 U are the amounts of amylosucrase in the treatment; [3] The particle diameter of the particle that 10%, 50%, and 90% of the cumulative sample mass; [4] Data are expressed as the mean value and standard deviation; [5] Values with different superscripts in each row are significantly different ($p < 0.05$), as determined by Duncan's multiple range test.

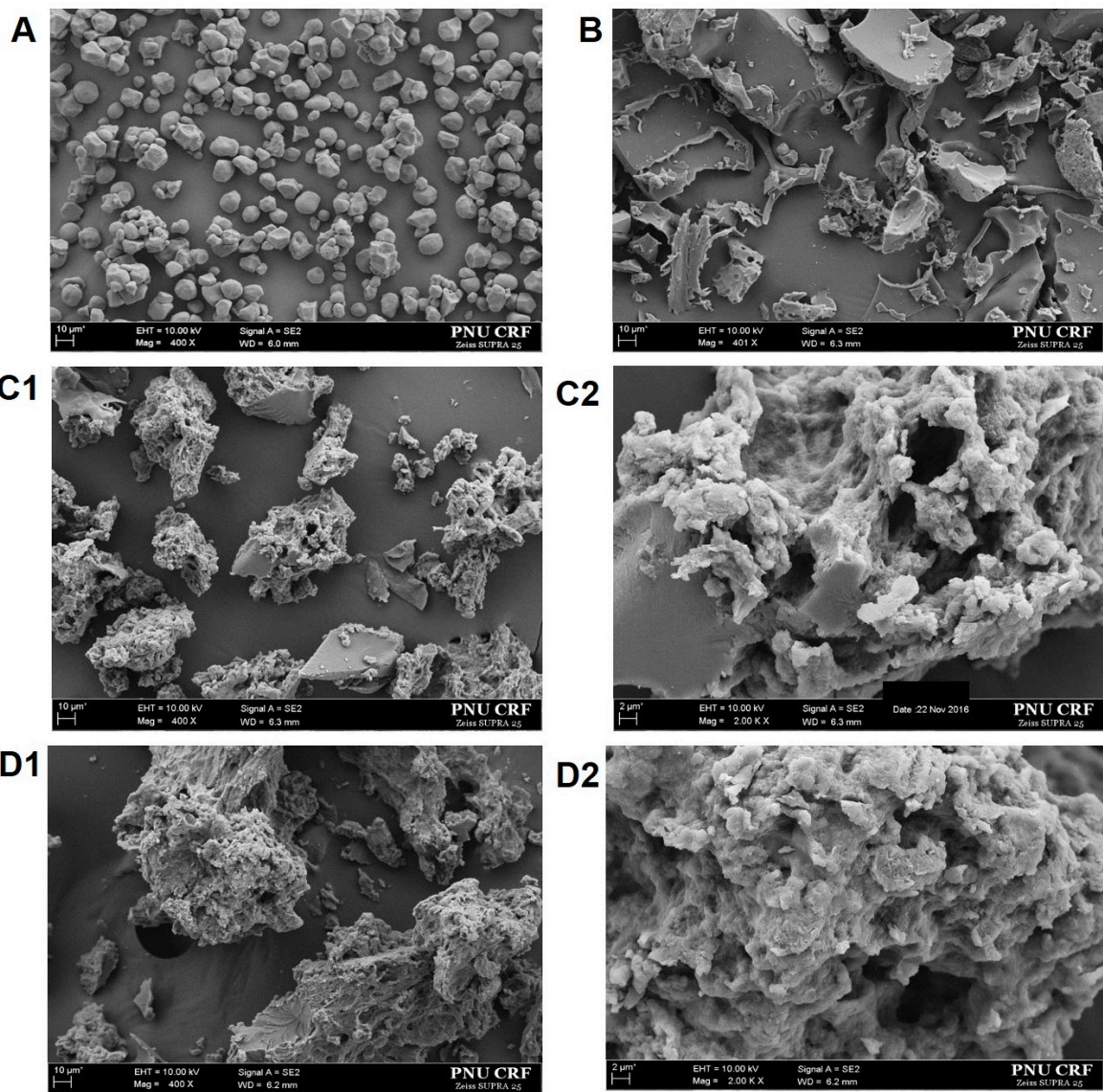

**Figure 1.** Field-emission scanning electron microscope images of (**A**) native waxy corn starch, (**B**) control starch, (**C1,C2**) 230 U-AS-treated starch, and (**D1,D2**) 460 U-AS-treated starch. Magnifications: (**A,B,C1,D1**), ×400, and (**C2,D2**), ×2000.

*3.2. Field-Emission Scanning Electron Microscopy (FE-SEM)*

The surface features of native, control, and AS-treated starches are shown in Figure 1, and the inner features of the AS-treated starches cross-sectioned with a 2-µm-thick razor blade are shown in Figure 2. Native waxy corn starch was observed as round and polyhedral-shaped granules with peripheral pores, as previously reported [20] (Figure 1A). The control AS starch was prepared by first gelatinizing the starch by heating. It was then incubated at 30 °C for 20 h without enzyme addition; during this process, starch retrogradation occurred [12,16]. The control starch thus showed both gelatinization and partial retrogradation properties. It lost its granular structure and shape as gelatinization caused the granule to swell, disrupting its structure and resulting in the loss of crystallinity. The granule structure was altered into irregularly shaped particles with a flat, very compact, laminated, and dense formation due to partial retrogradation (Figure 1B).

The AS-treated starch showed a completely different morphology compared to the native and control starches. It was observed as a filamentous algae- or coral-like, irregularly shaped porous network (Figure 1C,D). The 230 U-AS-treated starch had a flattened part on the surface of the particle.

This is probably because the enzyme did not react sufficiently with the starch due to the low enzyme concentration (Figure 1C1,C2). The 460 U-AS-treated starch did not have the flattened region on the granule surface and showed a well-formed coral-like shape, which was piled up layer by layer (Figure 1D1,D2). It is postulated that AS addition leads to glucose elongation at the ends of the starch residue, altering its surface to a porous and coral-like morphology. A previous study also mentioned that branch chains elongated by AS treatment retrograde more readily, resulting in the formation of more irregular-shaped particles than occur in control starch [17,21,22].

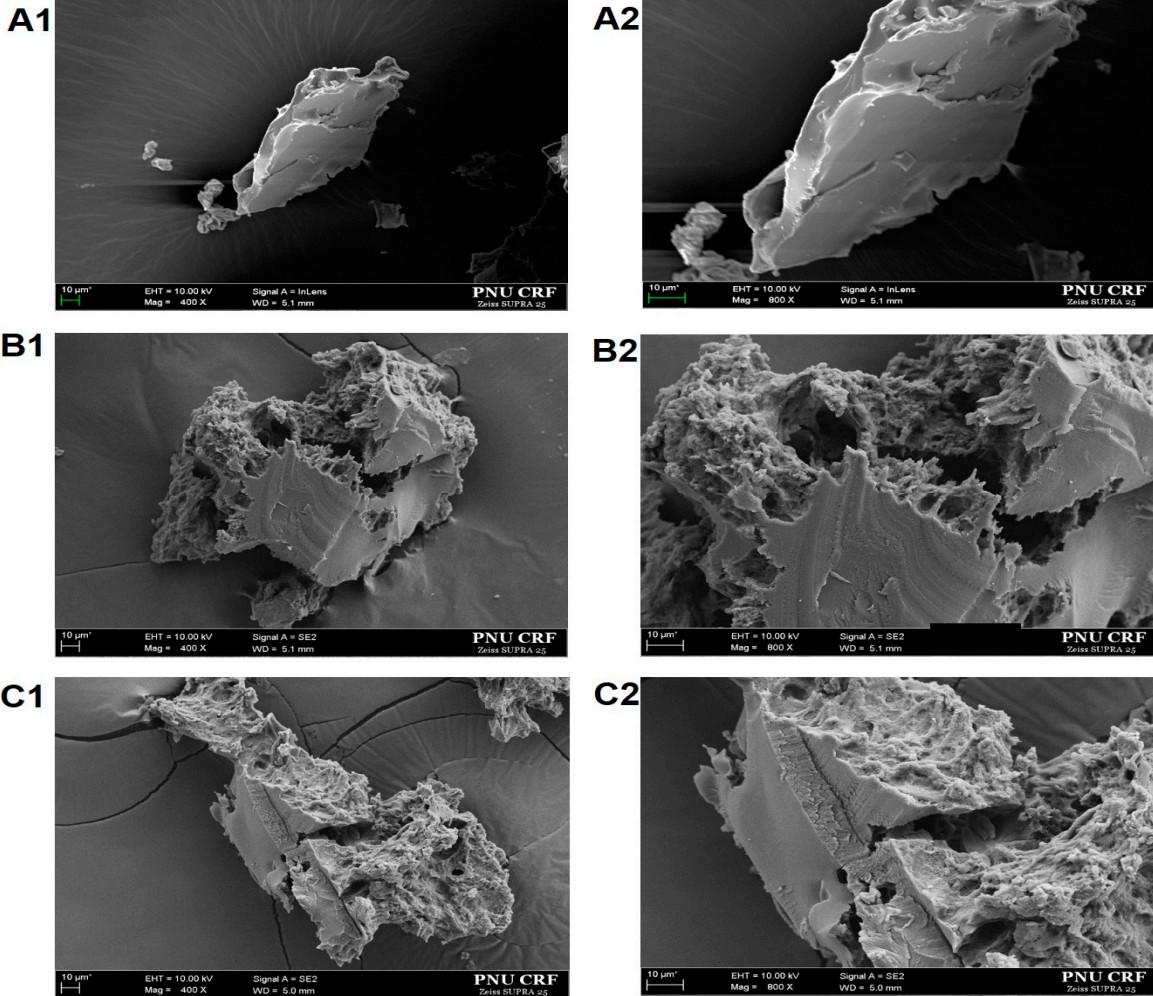

**Figure 2.** Field-emission scanning electron microscope images of the cross section of starch granules of (**A1**,**A2**) control starch, (**B1**,**B2**) 230 U-AS-treated starch, and (**C1**,**C2**) 460 U-AS-treated starch. Magnifications: (**A1**,**B1**,**C1**), ×400, and (**A2**,**B2**,**C2**), ×800.

Cross-sectional views of the control and AS-treated starch are shown in Figure 2. The cross-sectional view of control starch shows a very flat and dense shape (Figure 2A1,A2), while AS elongated the glucose using the gelatinized and retrograded starch as a primer, initiating glucose addition to the surface of the starch, producing a heterogeneous filamentous algae- or coral-like shape. The cross-sectional view of AS-treated starch revealed that the inner part of the starch was flattened and dense, indicating that the enzyme did not attach to or react with the inside of either the 230 U- (Figure 2B1,B2) or 460 U-AS-treated starch (Figure 2C1,C2). Glucose was observed to be elongated on the surface of the starch, as mentioned above.

Synthesis without a primer generated a more homogeneous and rounder product [6], but AS-treated starch showed a more heterogeneous shape. These phenomena agree with a previous

report by Zhang et al. [17] wherein elongated AS with branch chains of amylopectin acted like amylose, which is more easily retrograded, thereby forming an irregular shape.

Previous studies have shown that the structure of AS-treated starches is similar to that of enzyme-resistant starch produced by autoclaving–cooling cycling [23,24]. These studies showed that autoclaving–cooling cycling of amylomaize starch altered its porous and sponge-like network; further, four cycles of autoclaving–cooling yielded a more compact and porous structure than did one autoclaving–cooling cycle. Four cycles of autoclaving–cooling increased the RS content to 31.8%. This is similar to results obtained in the previous study. AS modification with 230 and 460 U AS increased the SDS fraction in starch by 30.3 and 42.7%, respectively. Furthermore, 460 U-treated starch had significantly higher RS content when compared to other samples, reaching 32.7% [10].

Although these studies used different techniques to modify starch (physical modification using autoclaving–cooling cycling versus enzymatic modification in the current study), both generated starches of similar morphologies and nutritional composition—compact, porous, sponge- or coral-like shapes with higher SDS and RS fractions than native starch. Further, RS residues isolated from autoclaving–cooling cycling of amylomaize starch after treatment with $\alpha$-amylase and amyloglucosidase. Isolated RS residues was no longer observed the porous structure, having been digested and removed by $\alpha$-amylase and amyloglucosidase treatment; this indicates that the porous part should be the RDS and SDS fraction, because it is digested by pancreatic amylase [14,23,24].

Starch modification with pullulanase and debranching enzyme under different reaction conditions resulted in porous and sponge-like starches similar to the AS-modified starch in this study [25,26]. AS, a member of the glycosyltransferase family, adds glucose and elongates it on a certain primer, while pullulanase, a hydrolase family enzyme, specifically attacks the $\alpha$-1,6 linkages and cleaves and debranches the starch [26,27]. The modes of action of AS and pullulanase are contrasting, but the morphology of the products was observed to be similar. On the other hand, starch pregelatinized at 80 °C and treated with $\alpha$-amylase and pullulanase showed a more dense and laminated morphology and increased RS rather than SDS content [28]. Similar morphological characteristics were observed in starches modified either enzymatically (e.g., AS-treated starch or debranching-enzyme-treated starch) or physically (e.g., autoclaving and cooling cycled starch), where both treatments yielded low-GI starch.

## 4. Conclusions

The particle size and morphological characteristics of starch are assumed to be important as they contribute to its nutritional characteristics (i.e., SDS or RS). AS reacted on the surface of starch rather than with the inner part; furthermore, bigger particle size and sponge-like morphology determined higher SDS and RS content. Further study is necessary to verify the relationship between amylolysis phenomena and particle size and microscopic characteristics.

**Author Contributions:** Conceptualization, I.P. and H.L.; methodology, I.P.; software, H.L.; validation, I.P. and H.L.; formal analysis, H.L.; investigation, I.P.; resources, I.P.; data curation, I.P.; writing—original draft preparation, I.P.; writing—review and editing H.L. visualization, H.L.; supervision, I.P.; project administration, I.P.; funding acquisition, I.P. All authors have read and agreed to the published version of the manuscript.

**Funding:** This work was supported by Research Fund 2020 through Youngsan University, Busan, Korea.

**Acknowledgments:** The Authors thank Cheon-Seok Park at Kyunghee University for assist NPAS production.

**Conflicts of Interest:** The authors declare no conflict of interest.

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
