# Peer review of "The Influence of Starch Modification with Amylosucrase Treatment on Morphological Features"

_processes, doi:10.3390/pr8111409_

Round 1
Reviewer 1 Report
Starch, which is a reserve material for plants, is an important source of energy in the daily diet. It is widely used in the food industry as a structure-forming ingredient in many dishes. Starch used in the food industry is commonly subjected to chemical, physical or enzymatic modifications, which results in greater stability in the product, better texture of gruels, resistance to acid or alkaline environment, and better solubility.
Due to the widespread use of modified starches in the food industry, I consider undertaking this research justified.
The material and methodology of the research have been sufficiently described, please only provide the ethanol concentration (lines 71, 72) and the number of rotations of the centrifuge (lines 73, 74).
The results were spoken and discussed with the literature data. Please provide numerical data with the same number of decimal places in the text (lines 105, 108, 109) and in table 1.
There is no summary in the article, please supplement it.
Reviewer 2 Report
The article is interesting and could be accepted for publication, but after minor adjustments.
Here is the list of the suggested corrections:
The title of the article ‘Morphological features of amylosucrase-treated starch increased in slowly digestible and resistant starch fractions’ suggests that the content of SDS and RS in AS-treated starch was determined. However these values were not marked in this article. The Authors discussed the influence of starch modification on the formation of RS and SDS fractions only on the basis of literature data. Therefore, I believe the title should be changed. For example ‘The influence of starch modification with amylosucrase on morphological features of the obtained preparations’ or ‘The morphological feature of starch after amylosucrase treatment’.
Line 152-155 the fragment describing the cross-section of starch granules, concerns Figure 2.- It appears from the text that figures B1 and B2 show the same preparation (230-AS- treated starch). Similarly Figure C1 and C2 (460-AS- treated starch). Nevertheless the images present different preparations. At the same time, C1 is an enlarged image of B1 as well as C2 is an enlarged image of B2. I think that the caption under the figures should be changed or the photos should be arranged in a different way for the information provided to be reliable.
Line 166 'This is similar to results obtained in the present study' This sentence suggests that the content of SDS in the obtained preparations was investigated in this experiment, which is not true. These data, as well as the data quoted in the next sentence of the article (line 167), refer to the authors' earlier work. I propose to replace the ‘present’ word with the ‘previous’.
Line 171 The caption of Figure 2 should be corrected, or the order of the photos should be changed.
References - add DOI numbers where missing
